# Social Inequities in the Impact of COVID-19 Lockdown Measures on the Mental Health of a Large Sample of the Colombian Population (PSY-COVID Study)

**DOI:** 10.3390/jcm10225297

**Published:** 2021-11-15

**Authors:** Juan P. Sanabria-Mazo, Bernardo Useche-Aldana, Pedro P. Ochoa, Diego F. Rojas-Gualdrón, Corel Mateo-Canedo, Meritxell Carmona-Cervelló, Neus Crespo-Puig, Clara Selva-Olid, Anna Muro, Jorge L. Méndez-Ulrich, Albert Feliu-Soler, Juan V. Luciano, Antoni Sanz

**Affiliations:** 1Institut de Recerca Sant Joan de Déu, 08830 Esplugues de Llobregat, Spain; 2Teaching, Research & Innovation Unit, Parc Sanitari Sant Joan de Déu, 08830 St. Boi de Llobregat, Spain; 3Department of Basic, Developmental and Educational Psychology, Autonomous University of Barcelona, 08193 Cerdanyola del Vallès, Spain; corel.mateo@uab.cat (C.M.-C.); meritxell.carmona@e-campus.uab.cat (M.C.-C.); neus.crespo@uab.cat (N.C.-P.); anna.muro@uab.cat (A.M.); 4Colombian College of Psychologists, Bogota 110221, Colombia; buseche53@hotmail.com (B.U.-A.); pedro.ochoa@colpsic.org.co (P.P.O.); 5School of Medicine, CES University, Medellin 050021, Colombia; dfrojas@ces.edu.co; 6Studies in Psychology and Educational Sciences, Open University of Catalonia, 08018 Barcelona, Spain; cselvao@uoc.edu; 7Department of Methods of Research and Diagnosis in Education, University of Barcelona, 08035 Barcelona, Spain; jordi.mendez@ub.edu; 8Department of Clinical and Health Psychology, Autonomous University of Barcelona, 08193 Cerdanyola del Vallès, Spain; albert.feliu@uab.cat

**Keywords:** COVID-19, lockdown, anxiety, depression, somatization, cross-sectional study

## Abstract

The identification of general population groups particularly vulnerable to the impact of COVID-19 lockdown measures on mental health and the development of healthcare policies are priority challenges in the current and future pandemics. This study aimed to identify the personal and social determinants of the impact of COVID-19 lockdown measures on mental health in a large sample of the Colombian population. In this cross-sectional study, an anonymous online survey was answered by 18,061 participants from the general population residing in Colombia during the first wave of the COVID-19 outbreak (from 20 May to 20 June 2020). The risk of depression, anxiety, and somatization disorders were measured using the Patient Health Questionnaire (PHQ-2), Generalized Anxiety Disorder Scale (GAD-2), and Somatic Symptom Questionnaire (SSQ-5), respectively. Overall, 35% of participants showed risk of depression, 29% of anxiety, and 31% of somatization. According to the analysis of social determinants of health, the most affected groups were people with low incomes, students, and young adults (18–29 years). Specifically, low-income young females were the most at-risk population group. These findings show how the lockdown measures affected the general population’s mental health in Colombia and highlight some social risk factors in health.

## 1. Introduction

The first case of the new coronavirus disease (COVID-19) was reported in Wuhan (China) in December 2019, and it has spread exponentially around the world [1]. The outbreak of COVID-19 was declared a public health emergency by the World Health Organization (WHO) on 30 January and a pandemic on 11 March 2020. Since that moment, the severe acute respiratory syndrome caused by the pathogen SARS-CoV-2 has had an intense and extensive impact on the general population, causing more than 4,975,000 deaths worldwide and 127,000 deaths in Colombia, according to official reports presented as of 1 November 2021 [2]. About two-thirds of the world’s population has been subjected to lockdown measures involving prolonged home confinement and social restriction measures, making it the most potentially stressful life event shared by the largest number of people in the last century [3,4,5].

A pandemic is defined as “an epidemic occurring worldwide or over a very wide area, crossing international boundaries and usually affecting a large number of people” [6] (pp. 1). For this reason, it is considered a social health phenomenon with its structure and dynamics [7]. The design and development of epidemiological models that consider the psychosocial variables of the different infection links’ processes are a priority for public health [8]. In 2005, the WHO developed a model of the social determinants of health, differentiating between structural determinants related to the macro-economic and political context; and intermediate determinants associated with living conditions and biological and psychosocial factors [9]. The social determinants of health that have been systematically studied are gender, socioeconomic factors, working conditions, education level, ethnicity, and stages in the life cycle [10].

There is a consensus on the negative impact that mobility restrictions and the direct or indirect exposure to COVID-19 have had on the general population’s mental health [3,11,12,13,14,15]. A narrative review recently identified generalized anxiety, mood disorders, and stress in the general population during the first wave of the COVID-19 outbreak [14]. Another review found psychological alterations caused by lockdown measures, limitations on social interactions, and interference with work activity [3]. Some studies have reported in the early stages of a pandemic the effects of isolation [16] as increased prevalence of anxiety [17,18], depression [19,20,21], and post-traumatic stress [22], as well as increased anger [23], fear, sadness, and insomnia [3], and a high risk of social exclusion and stigmatisation [17,20].

Regarding the impact of COVID-19 lockdown measures on the general population’s mental health, a meta-analysis comprising 12 studies reported a prevalence of 25% anxiety and 28% depression in a sample of 27,475 people [13]. In another meta-analysis including 62 studies, a prevalence of 33% anxiety and 28% depression was found in a sample of 162,639 people [11]. A systematic review of 19 studies showed a prevalence of psychological distress ranging from 34% to 38%, depression from 15% to 48%, post-traumatic stress from 7% to 54%, and anxiety from 6% to 51% in a sample of 93,569 people [15]. A population-based study involving 6882 people in 59 countries found that 25% showed symptoms of moderate to severe depression and 19% of anxiety [24].

Furthermore, sociodemographic characteristics, clinical conditions, exposure to COVID-19, strict lockdown measures, and lifestyle changes have affected mental health [24]. Several studies have consistently highlighted how women and young people are at greater risk of mental disorders [15,25,26,27]. First-line clinicians, the elderly, university students, the LGBTQ+ community, economically vulnerable people, rural community, foreigners, people with chronic diseases (e.g., systemic autoimmune diseases), and mental health patients have also been identified as one of the populations most affected by COVID-19 lockdown measures [28,29,30]. Low income, low educational levels, chronic diseases, and a history of previous mental health condition have also been presented as factors associated with increased severity of psychopathological symptoms [1,15]. Nevertheless, further research is needed to assess this pandemic’s scope in countries such as Colombia [31], where mental health infrastructure is less developed, and the impact might be more severe [12].

There is growing evidence that social inequities explain mortality rates in a wide range of diseases [12,32]. In the case of COVID-19, some studies suggest an increased risk of infection by SARS-CoV-2 among social groups characterised by vulnerability factors [33,34] and an increase in the lethality of the disease [35]. This evidence about the unequal impact of COVID-19 on vulnerable groups has led to the current emergency being defined as a syndemic [36], in which socioeconomic factors interact with biological factors, increasing personal vulnerability. The information collected so far indicates that the virus spreads and causes higher mortality rates among the most vulnerable communities, which, in turn, are victims of pre-existing social and economic inequities. This is the case of Colombia, where higher mortality rates for COVID-19 are associated with socioeconomic inequality in terms of low income, gender, ethnicity, and type of health insurance [37]. Even in developed countries with more robust health systems, social determinants have been identified in which the most vulnerable groups were most affected [38,39].

Colombia is a country of particular interest for the study of how inequities can make the most vulnerable groups experience the rigors of the COVID-19 syndemic with greater intensity. First, because it is one of the countries that had the greatest structural economic inequality as a factor of general vulnerability prior to the pandemic. Despite positive developments in recent decades [40], Colombia continues to show one of the highest Gini coefficient values (50.4) in the world [41]. Second, because it is one of the countries in the world most affected by the pandemic [42], and where the authorities adopted the strictest mobility restriction measures [43], such as personal confinement at home, without this having been accompanied by economic compensation measures for a high percentage of the population that survives within an informal economy. All of this may explain the poor results in terms of controlling the pandemic in this country [44], exacerbating the impact on the most vulnerable groups in critical aspects such as mental health.

Therefore, this study aimed to identify the personal and social determinants of the impact of COVID-19 lockdown measures on mental health in a large sample of the Colombian population, according to the social determinants of health model.

## 2. Materials and Methods

### 2.1. Study Design

PSY-COVID is a cross-sectional study based on disseminating an anonymous online survey (Google Forms^®^) coordinated from the Autonomous University of Barcelona, Spain. This study was carried out in 30 countries (13 in the Americas, 11 in Europe, three in Asia, and three in Africa) and involved more than 150 international researchers from 56 scientific institutions. Specifically, this article reports the results of the sample resident in Colombia during the lockdown measures.

### 2.2. Participants

A total sample of 18,833 participants answered the online survey in Colombia, but 772 were excluded from this study because they resided in other countries during the period of mobility restrictions adopted in Colombia. Finally, the sample was composed of a total of 18,061 participants who met the inclusion criteria from all regions of the country. Inclusion criteria were: (1) adults (≥18 years old) and (2) residents in Colombia during COVID-19 lockdown measures.

### 2.3. Procedure

The PSY-COVID survey was developed by the research group considering previous relevant literature about mental health. The measures and instruments selected in this study were validated by a panel of 30 international experts in clinical and health psychology and public health. After linguistic and content validation, the online survey’s usability test was also carried out to ensure understanding of the questions. The online survey administration in Colombia was conducted using the snowball method, and the time to complete the survey was approximately 15 min. Responses were accepted from 20 May to 20 June 2020. The survey was distributed mainly through social networks (Facebook, Instagram, WhatsApp, Twitter, etc.), media (newspapers, television, radio, etc.), and institutional contacts (universities, foundations, health organisations, etc.). 

This study was conducted during a period of partial easing of the COVID-19 lockdown measures adopted by the Colombian authorities. This particular period of time was chosen in order to analyse the cumulative impact of COVID-19 lockdown measures on the mental health of the sample. During this phase, which started on 25 March 2020, the Colombian population was allowed to leave their homes only to purchase essential supplies, access health services, go to work, and assist and care for dependents. Figure 1 displays the criteria established to define the moment of dissemination of the study in Colombia.

The online survey included a consent to participate in the study and statements on data protection laws. This study was approved by the Animal and Human Experimentation Ethics Committee of the Autonomous University of Barcelona (CEEAH-5197) and validated by the ethical committee of the Colombian Association of Psychologists (Colpsic), in addition to following the guidelines of the Helsinki Declaration.

### 2.4. Measures

#### 2.4.1. Socio-Demographic Characteristics

The socio-demographic information questionnaire was used to describe the gender, age, income level, work status, educational level, ethnic group, area of residence, and region of residence.

#### 2.4.2. Mental Health Outcomes

Risk of depression disorder. The Patient Health Questionnaire (PHQ-2) was used to measure symptoms of depression [45,46]. The short version contains two items with a 4-point Likert response format, where 0 corresponds to not at all and 3 to nearly every day. The total score of the PHQ-2 ranges from 0 to 6. Participants were considered at risk of depression if the PHQ-2 score was ≥3 [45]. This instrument showed adequate internal consistency (Cronbach’s α = 0.79) in this sample [47].

Risk of anxiety disorder. The Generalized Anxiety Disorder Scale (GAD-2) was used to measure symptoms of anxiety [45,46]. The short version contains two items with a 4-point Likert response format, where 0 corresponds to not at all and 3 to nearly every day. The total GAD-2 score ranges from 0 to 6. Participants were considered at risk of anxiety if the GAD-2 score was ≥3 [45]. This instrument showed adequate internal consistency (Cronbach’s α = 0.83) in this sample [47].

Risk of somatization disorder. The Somatic Symptom Questionnaire (SSQ-5) was used to measure somatic symptoms. It was developed by the authors of the PSY-COVID questionnaire considering the meta-analysis results by Zijlema et al. [48] on scales measuring somatization. This instrument was assessed with a 4-point Likert-type response format, where 0 corresponds to not at all and 3 to nearly every day. The total SSQ-5 score ranges from 0 to 15. Participants were considered at risk of anxiety if the SSQ-5 score was ≥4. This instrument showed a single factor solution in exploratory factor analysis and an adequate internal consistency (Cronbach’s α = 0.77) in this sample.

### 2.5. Data Analysis

The descriptive analysis of the sample’s characteristics was carried out using frequencies and percentages. For mental health outcomes, overall and subgroup prevalence (Prev.) were estimated. The association analyses were performed using the generalized linear model with binomial family and logarithmic link function considering each mental health outcome (i.e., risk of depression, anxiety, and somatization disorder) as the dependent variable and all socio-demographic characteristics (i.e., gender, age, income level, work status, educational level, ethnic group, area of residence, and region of residence) as independent variables. Simple and multiple regression (enter method) models were obtained to estimate observed and adjusted prevalence ratios (aPR) with 95% confidence intervals (95% CI), respectively. This modelling alternative was considered, as included socio-demographic characteristics are proxies of the individual level structural social determinants of health inequalities [10]. When convergence was not achieved for the logarithmic link function, the logit link function was used, and results were presented as observed and adjusted odds ratios (aOR). *p*-values < 0.05 were considered statistically significant. Since all the items were forced choice in this study, there was no missing data. Statistical analyses were conducted in STATA v16.1 (College Station, TX, USA), and data processing was conducted in IBM SPSS v26.0 (IBM Inc., Armonk, NY, USA).

## 3. Results

### 3.1. Characteristics of the Sample

In total, the sample consisted of 18,061 residents in Colombia during the COVID-19 lockdown measures. Of them, 74.7% were female, 52.2% were middle-aged adults, 61.9% reported medium incomes, 45.5% were formal workers during the lockdown, 19.5% were students, 89.6% had a university education, 91.4% did not identify themselves as part of any ethnic group, 91.8% were living in urban areas, and 51.9% lived in the Andean region. Socio-demographic information is displayed in Table 1.

### 3.2. Impact of COVID-19 Lockdown Measures on the Mental Health

#### 3.2.1. Prevalence

Overall, 35.0% of the sample showed a prevalence of risk for depression, 29.2% of risk for anxiety, and 31.2% of risk for somatization during COVID-19 lockdown measures. Table 2 presents the prevalence of mental health risk by socio-demographic characteristics. In addition, the scores (M and SD) of mental health outcomes for each socio-demographic characteristic are described in the Appendix A (see Appendix A for more details).

#### 3.2.2. Social Determinants of Health

The specific results of the generalized linear model for the social determinants of health associated with each dimension affecting mental health are shown in Table 3.

##### Gender

Participants of non-binary gender showed the highest significant prevalence of risk for depression (Prev. = 56.0%; aOR = 2.50; 95% CI = 1.54–4.06), anxiety (Prev. = 40.0%; aPR = 1.42; 95% CI = 1.06–1.90), and somatization (Prev. = 41.3%; aPR = 2.12; 95% CI = 1.61–2.78). Females reported higher risks of significant prevalence than males for depression (Prev. = 36.0%; aOR = 1.17; 95% CI = 1.08–1.26), anxiety (Prev. = 30.7%; aPR = 1.21; 95% CI = 1.15–1.28), and somatization (Prev. = 35.0%; aPR = 1.71; 95% CI = 1.60–1.82). The lowest prevalence of risk on mental disorders was identified in males.

##### Age Group

Young adults reported the highest significant prevalence of risk for depression (Prev. = 48.0%; aOR = 3.56; 95% IC = 2.93–4.31), and somatization (Prev. = 40.1%; aPR = 3.07; 95% CI = 2.52–3.74). Middle aged adults also showed significant higher risk prevalence of depression (Prev. = 26.0%; aOR = 1.55; 95% CI = 1.28–1.88), anxiety (Prev. = 23.7%; aPR = 1.25; CI 95% = 1.08–1.46), and somatization (Prev. = 25.8%; aPR = 2.13; 95% CI = 1.75–2.59) than older adults. No statistically significant differences were identified in older people on any of the mental health indicators.

##### Income Level

Participants with lower incomes showed the highest significant prevalence of risk for depression (Prev. = 46.1%; aOR = 1.76; 95% CI = 1.54–2.00), anxiety (Prev. = 36.2%; aPR = 1.23; 95% CI = 1.12–1.35), and somatization (Prev. = 35.5%; aPR = 1.26; 95% CI = 1.15–1.39). Similarly, participants with medium incomes reported higher prevalence of risk for depression (Prev. = 31.4%; aOR = 1.15; 95% CI = 1.02–1.30), and somatization (Prev. = 35.5%; aPR = 1.26; 95% CI = 1.15–1.39) than high incomes.

##### Work Status

Students obtained the highest significant prevalence of risk for depression (Prev. = 46.0%; aOR = 1.48; 95% CI = 1.31–1.67), anxiety (Prev. = 35.6%; aPR = 1.21; 95% CI = 1.03–1.42), and somatization (Prev. = 37.8%; aPR = 1.30; 95% CI = 1.10–1.54). Unpaid workers also showed a higher significant prevalence of risk for depression (Prev. = 36.7%; aOR = 1.33; 95% CI = 1.08–1.64), as did formal workers (Prev. = 32.4%; aOR = 1.12; 95% CI = 1.00–1.24), compared to others of different work status.

##### Education Level

Participants with no formal education showed the highest significant prevalence of risk for somatization (Prev. = 50.0%; aPR = 1.85; 95% CI = 1.19–2.89), followed by participants with a university-level education (Prev. = 31.8%; aPR = 1.23; 95% CI = 1.12–1.34). On the other hand, participants with a primary education reported significant prevalence of risk for anxiety (Prev. = 33.2%; aPR = 1.22; 95% CI = 1.04–1.44), as did participants with a secondary education (Prev. = 34.1%; aPR = 1.16; 95% CI = 1.07–1.25). 

##### Ethnic Group

No significant higher prevalence was found by ethnicity. The highest considerable prevalence of somatization risk was found in participants who did not identify with any ethnic group (Prev. = 31.5%; aPR = 1.16, 95% IC = 1.05–1.29). 

##### Area of Residence

Participants residing in urban areas presented higher significant prevalence of the risk for depression (Prev. = 35.2%; aOR = 1.23; 95% CI = 1.09–1.39), anxiety (Prev. = 29.4%; aPR = 1.12; 95% CI = 1.03–1.22), and somatization (Prev. = 31.6%; aPR = 1.23; 95% CI = 1.13–1.34) than participants residing in rural areas. 

##### Region

Participants residing in the Orinoco region (Prev. = 30.9%; aPR = 1.16, 95% CI = 1.00–1.35) and the Pacific region (Prev. = 31.0%; aPR = 1.13; 95% CI = 1.03–1.24) showed the highest prevalence of the risk for anxiety. No statistically significant differences were identified in the other mental health indicators.

## 4. Discussion

This cross-sectional study provides evidence of inequities in the mental health impact of the COVID-19 pandemic in the Colombian population during the first phase of lockdown measures. In line with other research conducted in the general population, it was found that about one-third of the sample showed risks of depression, anxiety, and somatization disorders [1,13,26,27,49]. Income level, occupation, age group, and gender were the social determinants of health with the most significant explanatory value for the impact on mental health. The population groups most affected by the pandemic were low income individuals, students, and young adults, with risk of depression between 46% and 56%, anxiety between 36% and 40%, and somatization between 38% and 42%, respectively. Specifically, low-income young females were the most at-risk population group. Area of residence was the variable with the weakest association, with educational level, ethnic group, and region in an intermediate position with regard to strength of association.

According to previous research, these findings highlight the unequal distribution of resources as a structural factor with a high impact on mental health [37,38,39]. In Colombia, there is evidence that mental health was precarious before the pandemic [31]. However, this study indicated that there were 2.5 to 2.8 times more people with risk of anxiety and 1.5 to 1.9 times more people with risk of depression in the first wave of the COVID-19 outbreak (from 20 May to 20 June 2020). Along the same lines, Caballero et al. [50] found that during a first lockdown period, low social capital, considered as a social determinant of health, is associated with a higher risk of depression, suicide, stress, and insomnia. The results obtained are similar compared to findings reported at the beginning of the pandemic [51] and at the end of the most severe mobility restrictions [52].

These findings also highlight the need to prioritise COVID-19 pandemic containment and mitigation interventions for low-income people, students, and young adults, as has been demonstrated in other research [28,29,30,53]. To overcome the social inequalities existing in Colombia, the main recommendations of these results are proposed in Table 4.

This study has some limitations and strengths that should be considered in the interpretation of the results. The non-probabilistic sampling of this study limits the generalization of findings. Additionally, online data collection led to the underrepresentation of population groups without internet connectivity, lower digital literacy, older adults, residents in rural areas, and lower education levels. This underrepresentation may imply lower statistical power for association analyses. Finally, the mental health indicators assessed came from self-report instruments, which may introduce interpretation biases according to socio-demographic characteristics. In terms of strengths, the sample size and socio-demographic heterogeneity allowed for robust response ratios in the different social determinants of health explored in this study’s analyses. Likewise, the instruments used, validated by a large panel of international experts, obtained adequate study sample reliability. Despite limitations, the socioeconomic factors identified as associated with the COVID-19 lockdown’s higher impact provided relevant evidence for Colombia’s mental health care prioritization.

## 5. Conclusions

This study provides evidence of the significant risk of depression, anxiety, and somatization in the Colombian population during the lockdown period. The analysis of social determinants of health pointed out that the most affected groups in terms of mental health during the COVID-19 lockdown were individuals of low income, students, and young adults. Specifically, low-income young females were the most at-risk population group. These findings highlight social inequities in mental health during the first phase of lockdown measures and the need to enhance and develop public health strategies to overcome these regarding mental health in countries such as Colombia. Moreover, the fight against the coronavirus seems to be more challenging for certain groups of people probably because of social inequities. In other words, we are confronting a syndemic (not a pandemic) given that pre-existing social inequities in interaction with biological factors are exacerbating the impact of COVID-19 in specific territories and groups of individuals [56]. Furthermore, COVID-19 is probably exacerbating social inequities, further harming health and destroying jobs mainly among specific vulnerable labour sectors of our societies. Our findings might be of potential interest for policymakers and regulators involved in the allocation of public resources.

## Figures and Tables

**Figure 1 jcm-10-05297-f001:**
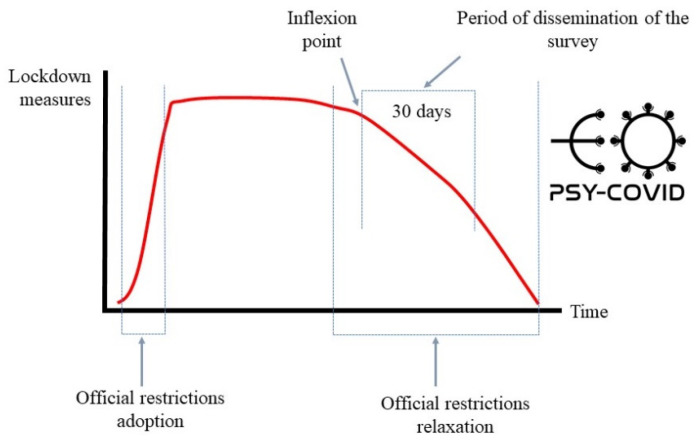
Selection of the time of dissemination of the survey.

**Table 1 jcm-10-05297-t001:** Socio-demographic characteristics.

Variables, *n* (%)	Sample (*n* = 18,061)
Gender	
Female	13,491 (74.7)
Male	4495 (24.9)
Non-binary	75 (0.4)
Age group	
Young adults (18–29 years)	7739 (42.9)
Middle aged adults (30–59 years)	9421 (52.1)
Older adults (≥60 years)	901 (5.0)
Income level	
Low	5173 (28.6)
Medium	11,186 (61.9)
High	1702 (9.5)
Work status	
Student	3511 (19.5)
Informal workers	2149 (11.9)
Formal workers	8208 (45.5)
Unpaid workers	515 (2.9)
Unemployed	3024 (16.8)
Retired	627 (3.4)
Education level	
No studies	18 (0.1)
Primary	322 (1.8)
Secondary	1536 (8.5)
University	16,185 (89.6)
Ethnic group	
Gypsy	26 (0.1)
Afrodescendant	953 (5.3)
Indigenous	578 (3.2)
None of the above	16,443 (91.4)
Area of residence	
Urban	16,527 (91.8)
Rural	1473 (8.2)
Region	
Amazon	285 (1.6)
Andean	9347 (51.9)
Caribbean	1395 (7.8)
Orinoco	557 (3.1)
Pacific	6420 (35.6)

Note. *n* = frequency, % = percentage.

**Table 2 jcm-10-05297-t002:** Prevalence of mental health risk by social determinants of health.

Variables, *n* (%)	Depression (PHQ-2)	Anxiety (GAD-2)	Somatization (SSQ-5)
Gender			
Female	36.0	30.7	35.0
Male	31.6	24.7	19.5
Non-binary	56.0	40.0	41.3
Age group			
Young adults (18–29 years)	48.0	37.3	40.1
Middle aged adults (30–59 years)	26.0	23.7	25.8
Older adults (≥60 years)	17.4	17.9	10.5
Income level			
Low	46.1	36.2	35.5
Medium	31.4	26.9	30.4
High	25.2	23.6	22.7
Work status			
Student	46.0	35.6	37.8
Informal workers	30.7	26.6	27.5
Formal workers	32.4	27.9	30.5
Unpaid workers	36.7	30.3	28.2
Unemployed	34.1	28.7	30.8
Retired	23.8	21.5	18.7
Education level			
No studies	38.9	44.4	50.0
Primary	37.3	33.2	25.8
Secondary	41.9	34.1	24.9
University	34.3	28.7	31.8
Ethnic group			
Gypsy	50.0	34.6	23.1
Afrodescendant	38.1	30.9	27.6
Indigenous	37.5	31.5	27.9
None of the above	34.7	29.0	31.5
Area of residence			
Urban	35.2	29.4	31.6
Rural	32.5	27.0	25.5
Region			
Amazon	35.8	29.1	34.4
Andean	32.2	28.2	31.4
Caribbean	34.0	27.1	31.8
Orinoco	38.2	30.9	32.0
Pacific	39.0	31.0	30.6

Note. PHQ-2 = Patient Health Questionnaire; GAD-2 = Generalized Anxiety Disorder Scale, SSQ-5 = Somatic Symptoms Questionnaire.

**Table 3 jcm-10-05297-t003:** Observed and adjusted associations of social determinants of health with mental health risks.

Variables	Depression (PHQ-2)		Anxiety (GAD-2)		Somatization (SSQ-5)	
	OR (95% CI)	aOR (95% CI)	PR (95% CI)	aPR (95% CI)	PR (95% CI)	aPR (95% CI)
Gender						
Female	**1.22 (1.13–1.31)**	**1.17 (1.08–1.26)**	**1.35 (1.25–1.46)**	**1.21 (1.15–1.28)**	**2.22 (2.04–2.41)**	**1.71 (1.60–1.82) **
Male	1.00	1.00	1.00	1.00	1.00	1.00
Non-binary	**2.76 (1.74–4.37)**	**2.5 (1.54–4.06)**	** 2.04 (1.28–3.25)**	**1.42 (1.06–1.90)**	**2.90 (1.82–4.62)**	**2.12 (1.61–2.78)**
Age group						
Young adults (18–29 years)	**4.37 (3.66–5.22)**	**3.56 (2.93–4.31)**	**2.73 (2.29–3.26)**	**1.84 (1.58–2.14)**	**5.67 (4.56–7.05)**	**3.07 (2.52–3.74)**
Middle aged adults (30–59 years)	**1.67 (1.39–1.99)**	**1.55 (1.28–1.88)**	**1.43 (1.20–1.71)**	**1.25 (1.08–1.46)**	**2.95 (2.38–3.67)**	**2.13 (1.75–2.59)**
Older adults (≥ 60 years)	1.00	1.00	1.00	1.00	1.00	1.00
Income level						
Low	** 2.54 (2.25–2.87) **	** 1.76 (1.54–2.00) **	** 1.84 (1.62–2.08) **	** 1.23 (1.12–1.35) **	** 1.87 (1.65–2.12) **	** 1.26 (1.15–1.39) **
Medium	** 1.36 (1.21–1.53) **	** 1.15 (1.02–1.30) **	** 1.19 (1.05–1.34) **	** 1.03 (0.94–1.12) **	** 1.49 (1.32–1.68) **	** 1.19 (1.09–1.30) **
High	1.00	1.00	1.00	1.00	1.00	1.00
Work status						
Student	** 1.92 (1.72–2.15) **	** 1.48 (1.31–1.67) **	** 2.01 (1.64–2.46) **	** 1.21 (1.03–1.42) **	** 2.65 (2.14–3.28) **	** 1.30 (1.10–1.54) **
Informal workers	1.00	1.00	** 1.32 (1.07–1.64) **	1.02 (0.87–1.21)	** 1.65 (1.32–2.06) **	1.08 (0.91–1.29)
Formal workers	1.08 (0.97–1.20)	** 1.12 (1.00–1.24) **	** 1.41 (1.16–1.71) **	1.09 (0.93–1.27)	** 1.91 (1.56–2.35) **	1.16 (0.98–1.36)
Unpaid workers	** 1.31 (1.07–1.60) **	** 1.33 (1.08–1.64) **	** 1.58 (1.21–2.07) **	1.16 (0.95–1.41)	** 1.71 (1.29–2.26) **	1.11 (0.90–1.36)
Unemployed	** 1.17 (1.04–1.32) **	1.03 (0.91–1.17)	** 1.46 (1.19–1.80) **	1.02 (0.87–1.21)	** 1.94 (1.56–2.40) **	1.11 (0.94–1.32)
Retired	0.70 (0.57–0.86)	1.05 (0.84–1.31)	1.00	1.00	1.00	1.00
Education level						
No studies	1.00	1.00	1.99 (0.78–5.04)	1.36 (0.82–2.25)	** 3.01 (1.19–7.64) **	** 1.85 (1.19–2.89) **
Primary	0.93 (0.35–2.47)	1.59 (0.59–4.33)	1.24 (0.98–1.56)	** 1.22 (1.04–1.44) **	1.05 (0.79–1.38)	1.18 (0.96–1.44)
Secondary	1.13 (0.44–2.93)	1.61 (0.60–4.28)	** 1.29 (1.15–1.44) **	** 1.16 (1.07–1.25) **	1.00	1.00
University	0.82 (0.32–2.12)	1.23 (0.46–3.27)	1.00	1.00	** 1.41 (1.25–1.59) **	** 1.23 (1.12–1.34) **
Ethnic group						
Gypsy	1.89 (0.87–4.07)	2.05 (0.90–4.67)	1.30 (0.58–2.91)	1.22 (0.73–2.04)	0.79 (0.31–1.98)	1.06 (0.54–2.08)
Afrodescendant	1.16 (1.01–1.33)	1.03 (0.89–1.18)	1.09 (0.95–1.26)	1.01 (0.92–1.12)	1.00	1.00
Indigenous	1.13 (0.96–1.35)	1.03 (0.86–1.23)	1.13 (0.94–1.35)	1.07 (0.95–1.20)	1.01 (0.80–1.28)	1.05 (0.89–1.23)
None of the above	1.00	1.00	1.00	1.00	** 1.20 (1.04–1.39) **	** 1.16 (1.05–1.29) **
Area of residence						
Urban	** 1.13 (1.01–1.26) **	** 1.23 (1.09–1.39) **	** 1.13 (1.01–1.27) **	** 1.12 (1.03–1.22) **	** 1.35 (1.20–1.53) **	** 1.23 (1.13–1.34) **
Rural	1.00	1.00	1.00	1.00	1.00	1.00
Region						
Amazon	1.00	1.00	1.11 (0.83–1.46)	1.03 (0.84–1.25)	1.19 (0.93–1.53)	1.08 (0.92–1.26)
Andean	0.85 (0.67–1.09)	1.03 (0.80–1.34)	1.06 (0.93–1.20)	1.08 (0.98–1.18)	1.04 (0.97–1.11)	1.02 (0.97–1.07)
Caribbean	0.92 (0.71–1.21)	1.05 (0.79–1.39)	1.00	1.00	1.06 (0.93–1.20)	1.01 (0.93–1.10)
Orinoco	1.11 (0.83–1.49)	1.33 (0.97–1.81)	1.20 (0.97–1.49)	** 1.16 (1.01–1.35) **	1.07 (0.89–1.29)	1.07 (0.95–1.21)
Pacific	1.15 (0.90–1.47)	1.25 (0.96–1.61)	** 1.21 (1.06–1.37) **	** 1.13 (1.03–1.24) **	1.00	1.00

Note. Prev. = Prevalence; OR = odds ratio; aPR = adjusted prevalence ratio; aOR = adjusted odds ratio; CI95% = 95% confidence interval. PHQ-2 = Patient Health Questionnaire; GAD-2 = Generalized Anxiety Disorder Scale; SSQ-5 = Somatic Symptoms Questionnaire. *p* Values < 0.05 are highlighted in bold.

**Table 4 jcm-10-05297-t004:** Main recommendations from study findings.

Main Recommendations
To develop health care intervention programs especially targeted at low-income young females (18–29 years), considering that they are the most at-risk population. To establish and maintain economic security measures for socio-economically disadvantaged population groups. To protect the unemployed and those in informal employment who are exposed to a higher risk of non-compliance with preventive behaviour. To formulate prevention, care, and mental health promotion interventions that target young, middle, and older adults differently, given that the pandemic’s impact is unequal among different sociodemographic groups. To implement specific mental health promotion actions for populations that do not identify with the binary classifications of male-female or masculine-feminine. To explore the impact of the COVID-19 pandemic on the mental health of highly vulnerable groups, such as patients with chronic diseases (e.g., systemic autoimmune diseases) and mental disorders [54,55].

## Data Availability

The data that support the findings of this study are available from the corresponding authors upon request.

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
