# Peer review of "Social Inequities in the Impact of COVID-19 Lockdown Measures on the Mental Health of a Large Sample of the Colombian Population (PSY-COVID Study)"

_jcm, 2021, doi:10.3390/jcm10225297_

Round 1

Reviewer 1 Report

The article its an interesting approach on the covid 19 pandemic specifically on the Colombian population. It is a now popular topic worldwide, but is nice to view how it had an impact on specific countries. The methodology and tools were applied in a simple form and well, so in my believe anyone who read it can understand it. 

Author Response

Thank you very much for your encouraging comments. We believe that this study provides strong support for the presence of social inequities in the impact of the COVID-19 pandemic on the mental health of the Colombian population.

Reviewer 2 Report

This study aimed to identify the personal and social determinants of the impact of COVID-19 lockdown measures on mental health in a large sample of the Colombian population, according to the social determinants of health model.

The study is well written and brings new insight into mental health in the time of the COVID-19 pandemic.
The study is of great value due to the size of the study group.
The Materials and Methods and Results sections do not need to be revised.
Nevertheless, in the introduction or discussion section, it is advisable to discuss the results of research relating to a very vulnerable group with regard to mental health in the COVID-19 pandemic, i.e. patients with autoimmune diseases.

Author Response

Thank you very much for your comments. We fully agree that it was necessary to highlight in the introduction (see adjustment on lines 84 to 96) and discussion (see adjustment on lines 312 to 314) the impact of the COVID-19 pandemic on the mental health of very vulnerable groups, such as patients with autoimmune diseases and people with mental disorders. We have included two references [54,55] that we consider relevant to support the evidence on the impact of the COVID-19 pandemic on the mental health of these specific population groups.

Reviewer 3 Report

Dear Authors,

I understand that the present study is focused on clarification of prevalence of mental health risk including depression, anxiety or somatization in Colombian population, as well as association between each mental health risk and social determinants. The findings in this nationwide survey are meaningful to develop effective strategies towards mental health risks during the Covid-19 pandemic. However, there are some information insufficient on data analysis using the log binomial multivariate modeling as indicated in the below comments. Thus, I kindly suggest that the manuscript needs to be rewritten considering additional explanation about data analyses.

Major comments

1. page 5, line 189; to statistically analyze the association between each mental health risk and social determinants, the log binomial multivariate model (is this binomial logistic regression analysis?) is applied in this study. To my knowledge regarding typical process of the modeling, at first, a dependent variable or independent variables (i.e. covariates) should be defined firstly in the section of Data analysis. In statistical methodology of this study, I guess that the dummy variable (such as non-depression group = 0, depressive symptoms’ group = 1) clarified by a cut-off value of 3 or more PHQ-2 scores was set up as a dependent variable for the binomial regression model analysis, but how a dependent variable (or a dummy variable) was defined for analysis should be clearly explained in the section of Methods.

2. Regarding a procedure of the binomial logistic regression modeling, typically, a selective method of independent variables to input into the regression model should be carried out according to advanced examinations of bivariate analysis (correlation analysis) or univariate analysis. In a statistical case of a forced entry method for regression modeling, the references related to targeted determinants (i.e. why these variables could be inputted into the regression model?) need to be cited or explained in the section of data analysis.

3. Although Table 1 list demographic data for the samples, demographic data of the groups classified according to a cut-off value of each mental health parameter (PHQ-2, GAD-2 or SSQ-5) should be demonstrated in table 1 or additional tables. In the above table, I believe that it would be better to indicate sample-size and mean (standard deviation, SD) for each group (e.g. non-depression group vs depression group according to a cut-off value of 3 or more PHQ-2 scores). An indication of these statistics will be essential before performing the binomial regression analysis, although the authors mention simply about prevalence (%) of each mental health risk in the section of “3.2.1. Prevalence”.

4. As the authors mentioned, samples are biased to young adults (18-29 years) and middle-aged adults (30-59 years), and a number of people aged 60 years or more (n =901) will be significantly smaller than others. As well as research limitations regarding a selective bias of persons without internet connectivity, the prevalence or regression model estimated in the current cross-sectional study should be interpreted considering the above selective bias (small sample-size in older adults). Also, the authors also need to refer to a bias of small samples for older adults.

Author Response

  1. page 5, line 189; to statistically analyze the association between each mental health risk and social determinants, the log binomial multivariate model (is this binomial logistic regression analysis?) is applied in this study. To my knowledge regarding typical process of the modeling, at first, a dependent variable or independent variables (i.e. covariates) should be defined firstly in the section of Data analysis. In statistical methodology of this study, I guess that the dummy variable (such as non-depression group = 0, depressive symptoms’ group = 1) clarified by a cut-off value of 3 or more PHQ-2 scores was set up as a dependent variable for the binomial regression model analysis, but how a dependent variable (or a dummy variable) was defined for analysis should be clearly explained in the section of Methods.

Authors

  • Thank you very much for your comments. In “2.4.2. Mental health outcomes” (see adjustment on lines 171 to 191) we clarified the operational definitions of risk of depression, anxiety, and somatization based on cut-off points for each instrument. Additionally, we modified “2.5. Data analysis” (see adjustment on lines 192 to 209) to clarify which variables were taken as dependent and which as independent variables.
  1. Regarding a procedure of the binomial logistic regression modeling, typically, a selective method of independent variables to input into the regression model should be carried out according to advanced examinations of bivariate analysis (correlation analysis) or univariate analysis. In a statistical case of a forced entry method for regression modeling, the references related to targeted determinants (i.e. why these variables could be inputted into the regression model?) need to be cited or explained in the section of data analysis.

Authors

  • We fully agree that there was a need for further clarification of the data analysis information in this study. The way in which we originally described the analysis procedure could be confusing for the reader. For this reason, we made explicit in “2.5. Data analysis” (see adjustment on lines 194 to 202) that all socio-demographic characteristics were forced as independent variables (enter method) and indicated the justification as they are proxies of individual structural determinants of health inequities. We also included observed estimations of association (simple regression) and added a note to this section. Thank you very much for helping us to clarify these possible confusions.
  1. Although Table 1 list demographic data for the samples, demographic data of the groups classified according to a cut-off value of each mental health parameter (PHQ-2, GAD-2 or SSQ-5) should be demonstrated in table 1 or additional tables. In the above table, I believe that it would be better to indicate sample-size and mean (standard deviation, SD) for each group (e.g. non-depression group vs depression group according to a cut-off value of 3 or more PHQ-2 scores). An indication of these statistics will be essential before performing the binomial regression analysis, although the authors mention simply about prevalence (%) of each mental health risk in the section of “3.2.1. Prevalence”.

Authors

  • We consider the suggestion to describe and compare socio-demographic characteristics of, for example, the depressed (cases) and non-depressed (controls) is the optimal descriptive analysis for case-control studies. Given that our study has a cross-sectional design, we present prevalences of mental health risks by socio-demographic characteristics (as opposed to prevalence of socio-demographic characteristics by mental health risk, the case-control analysis). However, we agree that unadjusted (i.e. bivariate) analysis is relevant for the reader and have therefore decided to include it. So we added Table 2 to present prevalences by socio-demographic characteristics (originally shown in “Table 2. Social determinants of health associated with mental health risk”) and modified “Table 2. Social determinants of health associated with mental health risk” to “Table 3. Observed and adjusted associations of social determinants of health associated with mental health risks”. The adjustments introduced in the manuscript are between lines 224 and 227 (Table 2) and 229 and 234 (Table 3). In addition, as requested, we have included a supplementary table (Table S1; see adjustment on lines 348 to 349) reporting mental health outcome scores (M and SD) for each socio-demographic characteristic. Thank you very much for your suggestion.
  1. As the authors mentioned, samples are biased to young adults (18-29 years) and middle-aged adults (30-59 years), and a number of people aged 60 years or more (n =901) will be significantly smaller than others. As well as research limitations regarding a selective bias of persons without internet connectivity, the prevalence or regression model estimated in the current cross-sectional study should be interpreted considering the above selective bias (small sample-size in older adults). Also, the authors also need to refer to a bias of small samples for older adults.

Authors

  • Thank you very much for your comment. We clarified that lower statistical power is the major potential implication of underrepresentation of specific groups (see adjustment on lines 320 to 321).

Round 2

Reviewer 3 Report

Dear Authors,

In the resubmitted manuscript, I confirm that the authors have revised the manuscript carefully considering reviewer's comments.